# Lysine benzoylation is a histone mark regulated by SIRT2

He Huang[1], Di Zhang[1], Yi Wang[2], Mathew Perez-Neut[1], Zhen Han[3], Y. George Zheng[3], Quan Hao[2] & Yingming Zhao[1]

Metabolic regulation of histone marks is associated with diverse biological processes through dynamically modulating chromatin structure and functions. Here we report the identification and characterization of a histone mark, lysine benzoylation ($K_{bz}$). Our study identifies 22 $K_{bz}$ sites on histones from HepG2 and RAW cells. This type of histone mark can be stimulated by sodium benzoate (SB), an FDA-approved drug and a widely used chemical food preservative, via generation of benzoyl CoA. By ChIP-seq and RNA-seq analysis, we demonstrate that histone $K_{bz}$ marks are associated with gene expression and have physiological relevance distinct from histone acetylation. In addition, we demonstrate that SIRT2, a $NAD^+$-dependent protein deacetylase, removes histone $K_{bz}$ both in vitro and in vivo. This study therefore reveals a new type of histone marks with potential physiological relevance and identifies possible non-canonical functions of a widely used chemical food preservative.

---

[1] Ben May Department for Cancer Research, The University of Chicago, Chicago, IL 60637, USA. [2] School of Biomedical Sciences, University of Hong Kong, Hong Kong, China. [3] Department of Pharmaceutical and Biomedical Sciences, University of Georgia, Athens, GA 30602, USA. Correspondence and requests for materials should be addressed to Y.Z. (email: Yingming.Zhao@uchicago.edu)

Chromatin structure and transcriptional activity of genes are regulated by diverse protein posttranslational modifications (PTMs) in histones (or histone marks)[1,2]. Accumulating evidence shows that the status of histone mark levels is coupled to cellular metabolism[3–6]. As an example, short-chain fatty acids can be produced by cellular metabolism or derived from fermentation from the gut microbiota. These metabolites can serve as precursors for generation of acyl-CoAs that can be used for histone acylations[7,8]. Some examples of these metabolites include crotonate, butyrate, α-ketoglutarate, and 3-hydroxybutyrate[9]. The existence of a variety of different acyl-CoAs raises the intriguing possibility of there being undescribed pathways by which cellular metabolism impacts epigenetics through metabolite-directed histone marks.

Benzoyl-CoA is a central intermediate in the degradation of a large number of aromatic growth substrates in bacteria and gut microflora[10]. In mammalian cells, a likely source of benzoyl-CoA is sodium benzoate (SB), one of the most commonly used preservatives worldwide with a maximum allowed concentration up to 0.1% in foods. SB has been successfully used as a drug for patients with acute hyperammonemia, a result of diverse urea cycle disorders[11]. Although SB is listed among the "generally regarded as safe" compounds by the United States Food and Drug Administration, emerging studies have indicated that exposure to SB may cause harm to consumers[12–14]. Furthermore, inappropriate doses of intravenous SB to patients have led to severe complications[15]. These findings suggest human health risks are associated with SB; however, the underlying biological mechanisms remain unknown.

Here we report the histone mark lysine benzoylation ($K_{bz}$). This histone mark has not been, to the best of our knowledge, previously described. We extensively characterize this new histone mark using a variety of chemical and biochemical methods, and detect 22 $K_{bz}$ sites located on histones in mammalian cells. Metabolic labeling experiments indicated that SB could stimulate histone $K_{bz}$ through the generation of cellular benzoyl CoA. Importantly, ChIP-seq (chromatin immunoprecipitation followed by sequencing) and RNA-seq results reveal that histone $K_{bz}$ is a mark enriched in gene promoters and has unique physiological relevance. We further show that SIRT2, a $NAD^+$-dependent protein deacetylase, can remove $K_{bz}$ both in vitro and in vivo. Therefore, this study discovers an epigenetic mechanism that potentially contributes physiological changes caused by a widely used food preservative.

## Results

**Identification and verification of histone $K_{bz}$.** To discover novel histone marks, we extracted core histones from HepG2 cells and performed tryptic digestion. The resulting tryptic histone peptides were subjected to high-performance liquid chromatography–tandem mass spectrometry (HPLC-MS/MS) analysis; the acquired MS/MS data were analyzed by the PTMap software with an unrestricted sequence alignment algorithm that enables searching mass shifts caused by PTMs[16]. Notably, a histone H2B peptide, PEPTK$_{+104.0268}$SAPAPK, was identified with a mass shift of +104.0268 Da at the lysine residue position H2BK5 (Fig. 1). This accurate mass shift was used to deduce the possible element compositions with a maximum allowance of 2 nitrogen atoms and a mass tolerance of ±0.02 Da[17]. Based on the deduction results, the formula $C_7H_4O$ is the most likely elemental composition, and only one reasonable chemical structure, benzoyl group, can be responsible for this formula (Fig. 1).

To confirm the deduced chemical structure of the mass shift, we synthesized a peptide that has the same sequence as the in vivo H2B peptide and contains a benzoyl group at H2BK5. MS/MS and coelution analysis were performed to compare the synthetic peptide with its in vivo counterpart (Fig. 2a, b). The results showed that the MS/MS spectra of two peptides matched very well and both peptides were coeluted. In addition, similar results were obtained from other two peptides, K$_{+104.0261}$STGGK$_{ac}$APR and K$_{+104.0260}$QLATK$_{ac}$AAR, from histone H3 in HepG2 cells (Supplementary Fig. 1). Theoretically, only chemically identical peptides can show the same MS/MS fragmentation patterns and HPLC retention times. Therefore, our MS/MS and coelution analysis confirmed that the mass shift of +104.0268 Da is caused by a new type of PTM, $K_{bz}$.

Next, to further confirm $K_{bz}$ and to detect in vivo $K_{bz}$ sites, we generated a pan-anti-$K_{bz}$ antibody. Pan anti-$K_{bz}$ specificity was evaluated by a dot blot assay (Fig. 2c), in which the pan-anti-$K_{bz}$ antibody can only recognize the peptides bearing a benzoyl lysine but not unmodified peptides or the peptides containing a broad spectrum of other PTMs, such as lysine acetylation ($K_{ac}$), propionylation, crotonylation ($K_{cr}$), tyrosine nitration, and serine/threonine phosphorylation. Using this antibody, we were able to detect $K_{bz}$ histones from HepG2 cells, mouse liver, and *Drosophila* S2 cells (Fig. 2d). $K_{bz}$ signals were detected among core histones from all the three species, indicating that $K_{bz}$ is an evolutionarily conserved histone mark in mammalian and insect cells.

**Mapping histone $K_{bz}$ sites in HepG2 and RAW cells.** To identify in vivo histone $K_{bz}$ marks, we carried out proteomic analysis using histone proteins extracted from HepG2 and RAW cells that were treated with 5 mM of SB. The extracted histones, with or without chemical propionylation, were tryptically digested and analyzed (see Online Methods). The spectra of the identified $K_{bz}$ peptides were manually verified to remove false positives. Using these procedures and criteria, we identified 22 unique histone $K_{bz}$ sites (Fig. 3, Supplementary Fig. 2, and Supplementary Table 1). The $K_{bz}$ sites in HepG2 cells distributed in a similar pattern to those found in RAW cells. Interestingly, the 22 $K_{bz}$ sites, either in HepG2 or in RAW cells, were mainly located on the N-terminal tails. In contrast, histone $K_{ac}$ and $K_{cr}$ sites are more widely spread, located in both N-terminal tails and other regions (Fig. 3). This line of evidence implies that $K_{bz}$ may have a different role from histone $K_{ac}$ and $K_{cr}$ in chromatin regulation.

**SB stimulates $K_{bz}$ by generating benzoyl CoA.** Acyl-CoAs serve as donors for acylation reactions, in which the acyl groups in acyl-CoA are transferred to the ε-nitrogen of lysine residues by diverse acyltransferases. Recent studies showed that short-chain fatty acids, such as sodium crotonate, sodium succinate, and sodium malonate, stimulate production of their corresponding acyl-CoAs in cells, in turn elevating levels of histone acylations[9,18,19]. $K_{bz}$ is chemically derived from SB; therefore, we speculate that SB may be used as a precursor to generate cellular benzoyl-CoA in vivo by a cellular lipid CoA synthetase, thereby stimulating histone $K_{bz}$ level. To test this hypothesis, we metabolically labeled HepG2 cells with isotopic $D_5$-SB for 24 h. Consistent with our hypothesis; $D_5$-benzoylation was detected in histone peptides. In addition, the mass spectrometry-based quantification experiment showed an increase of $D_5$-benzoyl-CoA in a dose-dependent manner upon $D_5$-SB treatment (Fig. 4a and Supplementary Fig. 3).

To further test whether SB could be used by cells for $K_{bz}$, we extracted histone proteins from the $D_5$-SB-treated HepG2 cells and performed chemical propionylation for an aliquot of the histones. Both the chemical propionylated and non-propionylated histones were then tryptically digested and analyzed. Our experiment detected 15 histone $K_{bz}$ sites bearing $D_5$ (Supplementary Fig. 4 and Supplementary Table 2), further

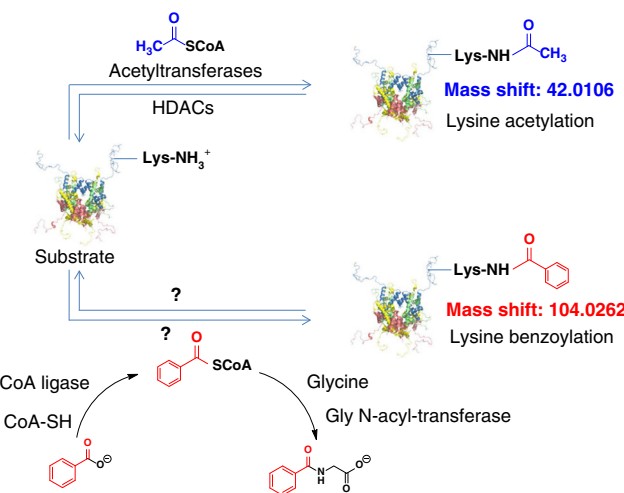

**Fig. 1** Lysine acetylation and benzoylation. Chemical structure of $K_{bz}$ that induces a mass shift of +104.0268 Da is shown in red

suggesting that SB is a precursor for the synthesis of benzoyl-CoA in the lysine benzoylation reaction.

**Dynamics of histone $K_{bz}$ in response to SB treatment.** To probe the dynamics of histone $K_{bz}$ exposed to SB, we experimentally increased benzoyl-CoA level in both HepG2 and RAW cells by adding SB. Western blot analysis showed significant enhancement of $K_{bz}$ on core histones after SB treatment, and we observed a dose-dependent increase in global histone $K_{bz}$ levels both in HepG2 and in RAW cells (Fig. 4b). In contrast, the global histone $K_{ac}$ levels slightly decreased in HepG2 cells and did not change in RAW cells upon SB treatment. These results were well confirmed by immunofluorescence staining, in which $K_{bz}$ levels substantially increased, whereas no obvious changes were observed for $K_{ac}$ levels (Fig. 4c).

To gain further insights into the dynamics of histone $K_{bz}$ stimulated by SB, we quantified the changes of histone $K_{bz}$ and $K_{ac}$ peptides in response to SB treatment in both HepG2 and RAW cells (Table 1). In this experiment, equal amounts of extracted histones from SB-treated and control cells were chemically labeled with $^{12}C$- or $^{13}C$-propionic anhydride, respectively. Then the labeled histones were mixed, tryptically digested, and analyzed by mass spectrometry. Ratios of SB treated to control for $K_{bz}$ and $K_{ac}$ peptides were calculated basing on their precursor intensities. To eliminate potential bias caused by protein expression changes, all the ratios of quantifiable $K_{bz}$ and $K_{ac}$ peptides were normalized by corresponding histone protein expression levels. The results indicated that 5 mM (~0.07%) of SB, a concentration lower than the maximum allowed percentage in food, dramatically increased the abundance of histone $K_{bz}$ (Table 1). For example, $H3K23_{bz}$ and $H4K8_{bz}$ sites in HepG2 cells, as well as $H4K5_{bz}$ and $H2AK13_{bz}$ sites in RAW cells, increased from 26.25- to 49.49-fold, while $H2AK9_{bz}$ in both HepG2 and RAW cells increased more than 50-fold. Interestingly, 2 $K_{bz}$ sites ($H4K12_{bz}$ and $H2AK13_{bz}$) in HepG2 cells and 2 $K_{bz}$ sites ($H3K14_{bz}$ and $H4K8_{bz}$) in RAW cells were detectable only in SB-treated cells. Consistent with the western blot analysis, the levels of quantifiable $K_{ac}$ sites in HepG2 cells decreased slightly, while the $K_{ac}$ sites in RAW cells were stable in response to SB treatment. Notably, most of the lysine residues bearing dynamic $K_{bz}$ modification are important to chromatin structure and function, suggesting important roles of $K_{bz}$ in the regulation of chromatin functions.

Given that $K_{ac}$ is the most abundant acylation in cells, next we compared the relative abundance of $K_{bz}$ with $K_{ac}$ by spectral counting. To this end, we performed immunoprecipitation experiments using histones extracted from SB-treated (5 mM for 24 h) HepG2 or RAW cells. Pan anti-$K_{ac}$ and pan anti-$K_{bz}$ antibodies were used to enrich $K_{ac}$ and $K_{bz}$ peptides. The results indicated that $K_{bz}$ levels on some sites are equivalent or close to corresponding $K_{ac}$ levels (Supplementary Fig. 5). For example, $K_{bz}$ levels on H2AK5, H2AK8, and H3K9 are as high as $K_{ac}$ levels, while $K_{bz}$ levels on H3K14 and H4K5 are higher than the half of corresponding $K_{ac}$ levels either in HepG2 or in RAW cells, indicating the high abundance of $K_{bz}$ in cells exposed to SB.

**SIRT2 removes histone $K_{bz}$ both in vitro and in vivo.** Histone deacetylases (HDACs) are a family of annotated enzymes that can remove acetyl groups from acetylated substrate proteins[20]. One of the HDACs, SIRT5, was identified as an $NAD^+$-dependent demalonylase, desuccinylase, and deglutarylase instead of deacetylase[18,21,22], suggesting that some HDACs may have non-canonical enzymatic activities and thus may work on histone $K_{bz}$.

To identify potential $K_{bz}$ deacylases, we screened all HDACs (including HDACs 1–11 and Sirtuin 1–7) using the synthetic peptide PEPTK$_{bz}$SAPAPK as a substrate. HPLC-MS/MS analysis showed that only SIRT2 exhibited significant lysine debenzoylation activity (Supplementary Fig. 6a). These results are consistent with our western blot assays in which core histone $K_{bz}$ levels of HepG2 and RAW cells did not visibly change following treatment with a class I/II HDAC inhibitor, Trichostatin A (TSA), even though the core histone $K_{ac}$ levels increased obviously under the same conditions (Supplementary Fig. 6b).

To confirm the direct debenzoylation activity of SIRT2, we carried out in vitro debenzoylation reactions using two additional synthetic peptides bearing benzoylated lysine residues ($K_{bz}$STGGK$_{ac}$APR and $K_{bz}$QLATK$_{ac}$AAR) as substrates (Fig. 5a). In both cases, we detected their corresponding unmodified counterparts. In contrast, the debenzoylated peptides were not detected in the absence of $NAD^+$, an essential cofactor for this reaction. Furthermore, this reaction was inhibited by the class III HDAC inhibitor nicotinamide but not by TSA. These results demonstrate that SIRT2 is a debenzoylase in vitro.

Given that SIRT2 is a known deacetylase, we next quantitatively compared its deacylase activity toward acetyl and benzoyl peptides. We performed kinetic studies using H2BK5$_{ac}$ or H2BK5$_{bz}$ peptides, respectively. The results showed that the $K_m$ for deacetylation was ~8.8-fold higher than that for debenzoylation, while the turnover number ($K_{cat}$) for deacetylation was ~57.1-fold higher than that for debenzoylation (Fig. 5b). Therefore, $K_{cat}/K_m$ for debenzoylation was approximately 1/6 of that for deacetylation.

We next examined whether SIRT2 has debenzoylase activity in vivo. By examining core histone $K_{bz}$ levels in $Sirt2^{+/+}$ (wild-type (WT)) and $Sirt2^{-/-}$ (knockout (KO)) mouse embryonic fibroblast (MEF) cells, we found that loss of SIRT2 expression was associated with increased $K_{bz}$ levels, especially on H3 and H2B proteins (Fig. 5c). In contrast, overexpression of SIRT2 in HEK293T cells by transient transfection led to decreased global $K_{bz}$ levels (Fig. 5d). Interestingly, these effects were significantly amplified by SB stimulation, while the levels of the H4K16$_{ac}$ positive control were unchanged following SB stimulation[23]. Collectively, these data indicate that SIRT2 is a debenzoylase both in vitro and in vivo.

To further understand the debenzoylase activity of SIRT2, we performed in silico molecular modeling to investigate the binding interaction between a benzoylated substrate and SIRT2. The

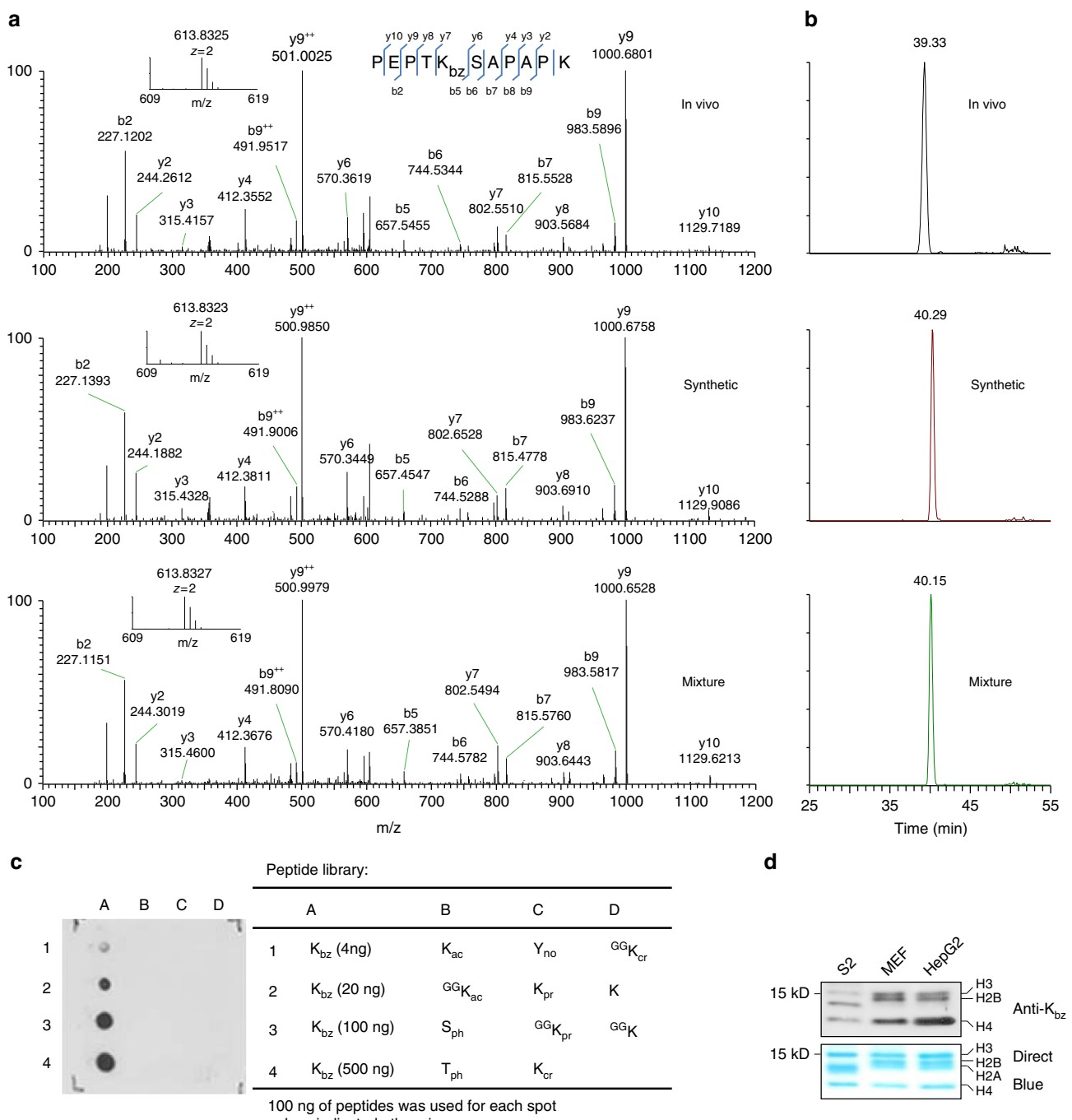

**Fig. 2** Identification and verification of histone $K_{bz}$. **a** The MS/MS spectra of an in vivo peptide bearing a PTM (PEPTK$_{+104.0268}$SAPAPK) (top), a synthetic lysine benzoylated peptide corresponding to the sequence of the in vivo peptide (middle), and a mixture of the two peptides (bottom). **b** Extracted ion chromatograms of the in vivo-derived peptide (PEPTK$_{+104.0268}$SAPAPK) (top), the synthetic $K_{bz}$ counterpart (middle), and their mixture (bottom) by HPLC-MS/MS analysis. **c** Dot blot assay of the pan anti-$K_{bz}$ antibody. Peptide libraries were used for the assay. Each peptide library contains 10 residues CXXXXKXXXX, where X is a mixture of 19 amino acids (excluding cysteine), C is cysteine, and the sixth residue is a modified lysine residue as indicated. **d** Detection of $K_{bz}$ in core histones from *Drosophila* S2 cells, MEF cells, and HepG2 cells

modeling results showed that benzoyl group on lysine side chain reached to the catalysis pocket center (Supplementary Fig. 6c). Similar to myristoyl substrate[24], the extensive hydrophobic interactions between benzoyl substrate and SIRT2 led to stronger binding energy than corresponding acetyl substrate (Supplementary Fig. 6d) and may account for much lower $K_m$ value of SIRT2 toward myristoyl or benzoyl peptides than acetyl peptide.

**Genome-wide mapping of histone $K_{bz}$ in HepG2 cells**. In order to investigate the genome-wide distribution of histone $K_{bz}$ mark, we carried out ChIP-seq with the pan anti-$K_{bz}$ antibody in HepG2 cells, also including the pan anti-$K_{ac}$ antibody as a control. In total, we identified 20283 $K_{bz}$ peaks distributed in 13,255 genes in SB-treated HepG2 cells. Analysis of the ChIP-seq datasets revealed that more than half (57.33%) of $K_{bz}$

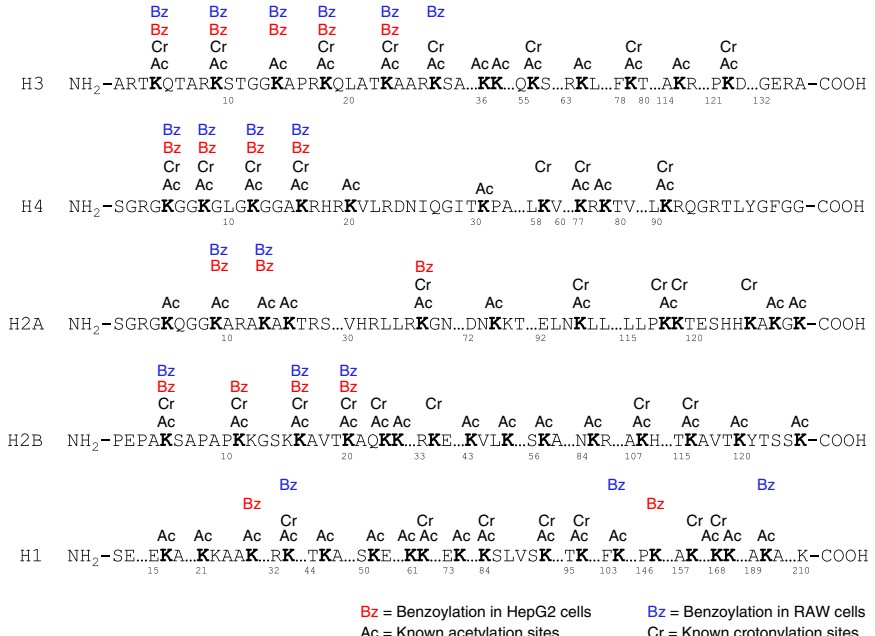

**Fig. 3** Illustrations of histone $K_{bz}$ in mammalian cells. The detected $K_{bz}$ sites in HepG2 and RAW cells are shown in red and blue, respectively. For comparison, the known $K_{ac}$ and $K_{cr}$ sites described in the literature are also listed

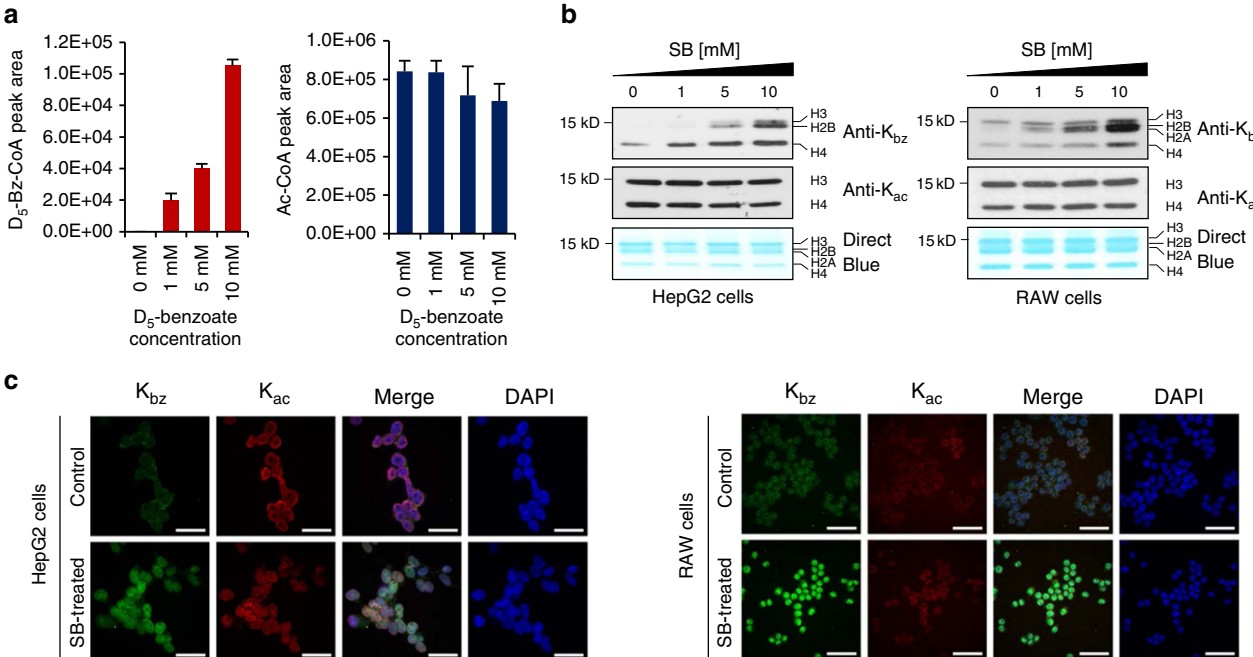

**Fig. 4** Dynamics of histone $K_{bz}$ in eukaryotic cells. **a** HPLC-MS/MS analysis of cellular benzoyl-CoA levels extracted from HepG2 cells cultured with the indicated concentration of $D_5$-SB (pH 7.4) for 24 h ($n = 3$, values are expressed as mean ± s.d.). **b** Western blotting analysis of core histone $K_{bz}$ in response to SB with the indicated concentration in HepG2 (left) and RAW (right) cells. **c** The dynamics of $K_{bz}$ and $K_{ac}$ in response to SB treatment as determined by immunofluorescence staining using pan-$K_{bz}$ and pan-$K_{ac}$ antibodies. HepG2 (left) and RAW (right) cells were treated with 10 mM of SB for 24 h. Scale bar, 50 μm

signals were associated with gene promoters (defined as ±3 kb around the transcription start sites [TSSs]; Fig. 6a). Similar to $K_{ac}$, $K_{bz}$ shows depletion in TSSs but is enriched around TSS regions, with slightly higher signals in upstream compared to downstream (Fig. 6b). These lines of evidence suggest that histone $K_{bz}$ may be associated with transcription.

We therefore examined the association between histone $K_{bz}$ mark and gene expression. To this end, we classified all genes into four categories based on their expression levels and plotted ChIP-seq peak

signals of $K_{bz}$ on these genes. Interestingly, we observed a positive correlation between $K_{bz}$ levels and gene expression at TSSs (Fig. 6c), which strongly support a role of histone $K_{bz}$ in gene expression.

**Profiling physiological relevance of histone $K_{bz}$ mark**. To explore the epigenetic role of histone $K_{bz}$, we first analyzed RNA-seq datasets from control and SB-treated HepG2 cells. In this analysis, quantified genes (false discovery rate (FDR) < 0.05) with

**Table 1 Quantified core histone $K_{bz}$ and $K_{ac}$ site dynamics in response to SB**

| Cell line | Site | Modified sequence | Ratio (T/C) |
|---|---|---|---|
| HepG2 | H3K23$_{bz}$ | K$_{pr}$QLATK$_{bz}$AAR | 27.38 |
| HepG2 | H4K8$_{bz}$ | GK$_{pr}$GGK$_{bz}$GLGK$_{ac}$GGAK$_{ac}$R | 26.35 |
| HepG2 | H4K12$_{bz}$ | GK$_{pr}$GGK$_{pr}$GLGK$_{bz}$GGAK$_{ac}$R | Treated only |
| HepG2 | H2AK9$_{bz}$ | GK$_{pr}$QGGK$_{bz}$AR | 57.67 |
| HepG2 | H2AK13$_{bz}$ | AK$_{bz}$AK$_{pr}$TR | Treated only |
| HepG2 | H3K27$_{ac}$ | K$_{ac}$SAPATGGVK$_{pr}$K$_{pr}$PHR | 0.75 |
| HepG2 | H4K8$_{ac}$ H4K12$_{ac}$ H4K16$_{ac}$ | GK$_{pr}$GGK$_{ac}$GLGK$_{ac}$GGAK$_{ac}$R | 0.66 |
| HepG2 | H2AK5$_{ac}$ | GK$_{ac}$QGGK$_{pr}$AR | 0.82 |
| RAW | H3K14$_{bz}$ | K$_{pr}$STGGK$_{bz}$APR | Treated only |
| RAW | H4K5$_{bz}$ | GK$_{bz}$GGK$_{pr}$GLGK$_{ac}$GGAK$_{ac}$R | 26.99 |
| RAW | H4K8$_{bz}$ | GK$_{pr}$GGK$_{bz}$GLGK$_{pr}$GGAK$_{ac}$R | Treated only |
| RAW | H2AK9$_{bz}$ | GK$_{pr}$QGGK$_{bz}$AR | 71.90 |
| RAW | H2AK13$_{bz}$ | AK$_{bz}$AK$_{pr}$SR | 49.69 |
| RAW | H3K27$_{ac}$ | K$_{ac}$SAPATGGVK$_{pr}$K$_{pr}$PHR | 1.01 |
| | | K$_{ac}$SAPSTGGVK$_{pr}$K$_{pr}$PHR | 1.17 |
| RAW | H2AK5$_{ac}$ | GK$_{ac}$QGGK$_{pr}$AR | 1.01 |
| | | GK$_{ac}$TGGK$_{pr}$AR | 0.87 |

The cells were untreated (C) or treated (T) with SB (5 mM, pH 7.4) for 24 h. Extracted histones from untreated and treated cells were chemically propionylated by [13]C- and [12]C-propionic anhydride, respectively. Ratios were normalized by the corresponding protein expression levels

the presence of $K_{bz}$ ChIP-seq peaks (FDR < 0.01 using magnetic-activated cell sorting (MACS)) in TSSs were selected. Then gene set enrichment analysis (GSEA) was used to compare the rank-ordered dataset of SB-treated versus control transcripts with respect to the KEGG pathways (Supplementary Data 1). The results showed that elevated $K_{bz}$ is positively correlated with glycerophospholipid metabolism, ovarian steroidogenesis, and phospholipase D signaling pathways (Fig. 6d).

To examine whether the upregulated genes in these pathways were regulated by $K_{bz}$ but not $K_{ac}$, we performed an additional KEGG pathway enrichment analysis using the ChIP-seq datasets (Supplementary Data 2). In this analysis, genes were selected based on three restrictive criteria: (1) the presence of $K_{bz}$ peaks (FDR < 0.01 using MACS) in TSSs with increased ChIP-seq read counts after SB treatment; (2) the presence of $K_{ac}$ peaks (FDR < 0.01 using MACS) in TSSs with decreased or unchanged ChIP-seq read counts after SB treatment; and (3) transcripts upregulated (FDR < 0.05 and at least 0.5 absolute log2 fold change) after SB treatment. This analysis identifies five pathways ($q < 0.05$, Fig. 6e): phospholipase D signaling, glycerophospholipid metabolism, ovarian steroidogenesis, serotonergic synapse, and insulin secretion. Among the five pathways, three were also identified in the RNA-seq GSEA analysis. These results confirmed that histone $K_{bz}$ marks are specifically located in chromatin regions that are associated with gene expression.

Next, we choose four genes to validate the correlation between histone $K_{bz}$ and gene expression by reverse transcriptase quantitative polymerase chain reaction (RT-qPCR) and ChIP-qPCR. The selected genes for validation were derived from the phospholipase D signaling, glycerophospholipid metabolism, ovarian steroidogenesis, and serotonergic synapse pathways, which were affected by increased $K_{bz}$ levels. These four genes are *PLA2G4C*, *ACHE*, *PLA2G15*, and *PTGS2*. A representative snapshot of the normalized ChIP-seq reads of selected genes *ACHE* and *PTGS2* is shown in Fig. 6f. RT-qPCR results showed

that expression of all the selected genes, which were marked by increased $K_{bz}$ but decreased $K_{ac}$, were upregulated with 1.28- to 1.55-fold increase (*PLA2G4C*: 1.44-fold, *ACHE*: 1.29-fold, *PLA2G15*: 1.28-fold, and *PTGS2*: 1.55-fold) (Fig. 6g). ChIP-qPCR results confirmed that the $K_{bz}$ levels at the promoters of selected genes increased after SB treatment, while the corresponding $K_{ac}$ levels decreased (Fig. 6h), indicating that the upregulation of these genes were associated with $K_{bz}$ instead of $K_{ac}$. Taken together, these data confirmed the correlation between histone $K_{bz}$ dynamics and gene expression, supporting potential physiological relevance of histone $K_{bz}$ mark and its role in the regulation of gene activity.

## Discussion

Emerging evidence suggests that short-chain histone lysine acylations are physiologically relevant and contribute to the regulation of chromatin structure and gene expression[1,17,25]. Despite the structural similarity, they are regulated by very different metabolic pathways and associated with unique physiology and diseases. $K_{bz}$ is structurally very different from the other short-chain lysine acylations. To date, $K_{bz}$ is, to the best of our knowledge, the only known histone lysine PTM bearing an aromatic acyl group. Compared with other histone marks bearing the short-chain fatty acids, $K_{bz}$ has lager molecular volume and stronger hydrophobicity (Supplementary Table 3). For example, the benzoyl group is more than two times the size of acetyl group, and the predicted log $P$ (logarithm of octanol–water partition coefficient) of benzoyl group is about three-fold higher than that of acetyl group. In eukaryotes, histone acylations constitute an important epigenetic mechanism that regulates gene expression patterns by loosening or tightening the interaction between DNA and histones. Therefore, we anticipate that histone $K_{bz}$ might have a more significant structural impact than $K_{ac}$ and lysine methylation on chromatin structure. In addition, due to the significant structural difference between $K_{bz}$ and other histone marks, it is tantalizing to speculate the possible existence of specific proteins or domains that recognize $K_{bz}$ and cause diverse downstream alterations in chromatin, providing an additional layer of control over gene transcription[26]. Furthermore, like other lysine acylations, $K_{bz}$ mark may be spread widely throughout the proteome and have important functions on non-histone proteins as well.

Our preliminary studies show that histone $K_{bz}$ has unique regulatory enzyme profiles. Histone $K_{ac}$ and a few other short-chain lysine acylations can be removed by multiple HDACs, such as HDAC1–3, HDAC6, and SIRT1–3[27,28]. In contrast, SIRT2 is the only HDAC that can remove $K_{bz}$ in our in vitro screen. These findings suggest the $K_{bz}$ pathway has a unique regulatory mechanism. Given the epigenetic roles of SIRT2 in diverse cellular processes[29–31], this study not only opens an opportunity for revealing unknown cellular mechanisms controlled by SIRT2 but also sheds some light on the epigenetic roles of $K_{bz}$.

The ChIP-seq and RNA-seq analyses revealed that the histone $K_{bz}$ mark, rather than the well-known $K_{ac}$, is involved in the expression of specific genes associated with glycerophospholipid metabolism, ovarian steroidogenesis, and phospholipase D signaling pathways. Thus histone $K_{bz}$ and $K_{ac}$ marks can be associated with distinct sets of genes. It has been reported that some lysine acylations, such as $K_{ac}$ and butyrylation ($K_{bu}$), or $K_{ac}$ and 2-hydroxyisobutyrylation ($K_{hib}$), can co-localize in the same genomic regions. In addition, the co-localization, such as histone H4 K5K8 acetylation and butyrylation, can act synergistically to enhance transcription activity in certain biological processes, such as sperm cell differentiation[32]. Therefore, the molecular mechanisms by which histone $K_{bz}$ exerts its functions are likely

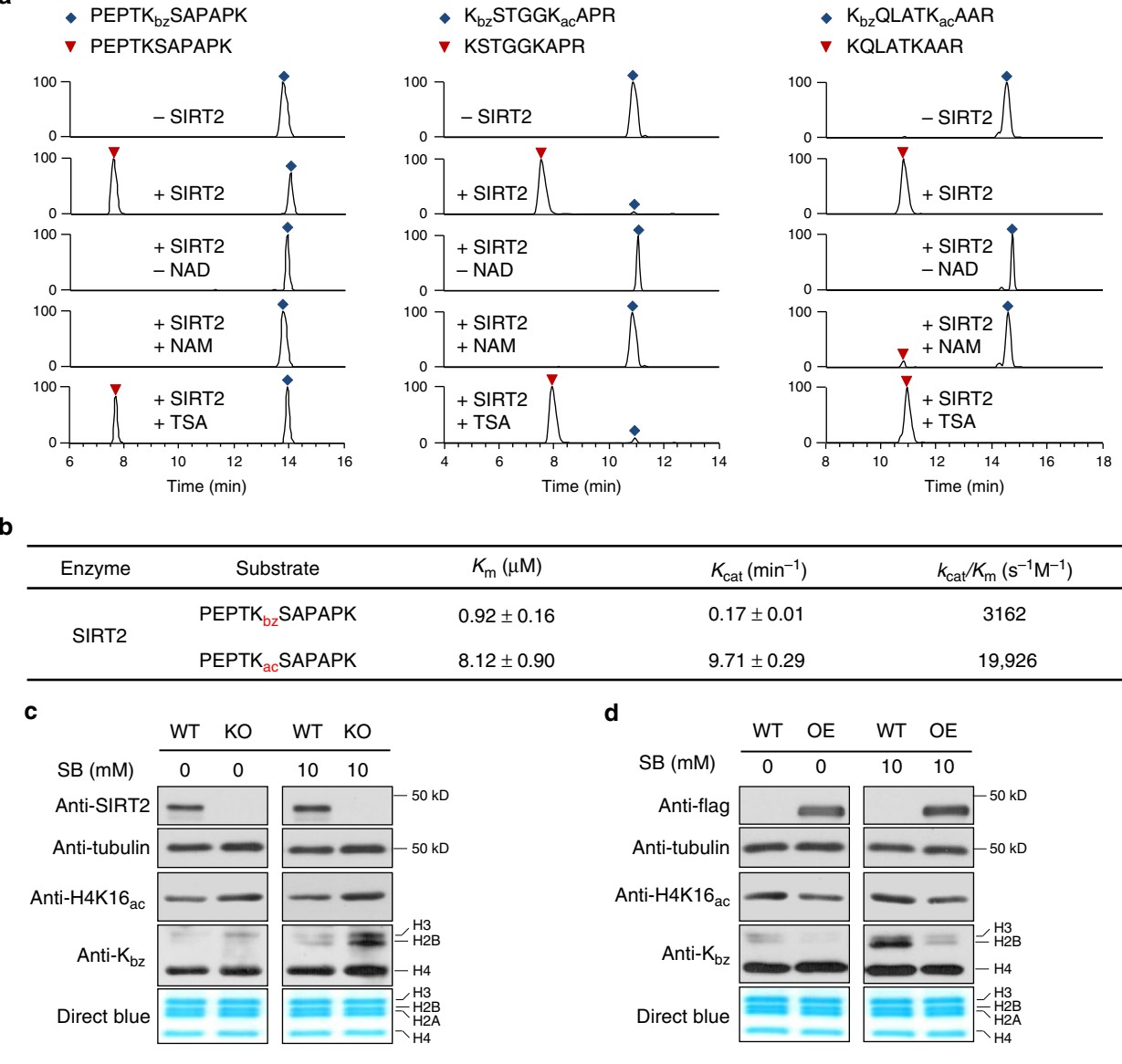

**Fig. 5** SIRT2 catalyzes histone lysine de-benzoylation both in vitro and in vivo. **a** SIRT2 can catalyze in vitro debenzoylation reactions of 3 synthetic $K_{bz}$ peptide substrates. **b** Kinetics data for SIRT2 on acetyl and benzoyl histone H2AK5 peptides. **c** Core histone $K_{bz}$ increased in SIRT2 knockout (KO) MEF cells. Whole-cell extracts were probed for the presence of SIRT2 and tubulin by western blot. Acid-extracted histones were tested for H4K16$_{ac}$ and global $K_{bz}$ levels by western blot. Total histones levels were visualized with Coomassie blue staining. **d** Core histone lysine benzoylation decreased in SIRT2 overexpressing (OE) cells. Flag-SIRT2 was transfected into HEK293T cells for 48 h. Whole-cell extracts were probed for the presence of Flag-SIRT2 and tubulin by western blot. Acid-extracted histones were tested for H4K16$_{ac}$ and global $K_{bz}$ levels by western blot. Total histones levels were visualized with Coomassie blue staining

different from those utilized by histone $K_{bu}$ and $K_{hib}$ sites in transcriptional regulation, although a clear physiological role for this modification still needs to be established in further studies.

Finally, we demonstrate that 5 mM (~0.07%) of SB can significantly increase histone $K_{bz}$ levels, which could pose potential safety concerns, as this concentration is lower than the maximum allowed percentage (0.1%) in food. Human plasma SB concentration can be as elevated as ~10 mM in patients with hyperammonemia receiving high doses of intravenous SB, which resulted in severe complications and showed toxicity of SB[15]. In addition, oral administration of high-dose SB increases the plasma SB concentration in healthy male volunteers in the range of 2–4 mM[33]. These concentrations of SB in plasma are similar to or even exceed the concentrations that we used for cell treatment

and could be associated with transcriptional responses. Moreover, it is becoming increasingly accepted that acyl-CoAs can be generated at local intracellular compartments, directly contributing to the PTM of proximal substrates[34]. Therefore, the concentration of benzoyl-CoA generated in proximity to chromatin could exceed the average cellular concentrations and could facilitate the addition of $K_{bz}$ to histones. These data suggest that the $K_{bz}$ mark could exist at physiologically relevant levels in cells. In fact, several studies in the past decade have suggested that SB could cause harmful effect on human health[12–14]. However, the underlying biological mechanism for SB function remains unclear. Thus our study not only discovers an epigenetic and PTM pathway but also sheds light on potential mechanisms for the physiological changes induced by SB.

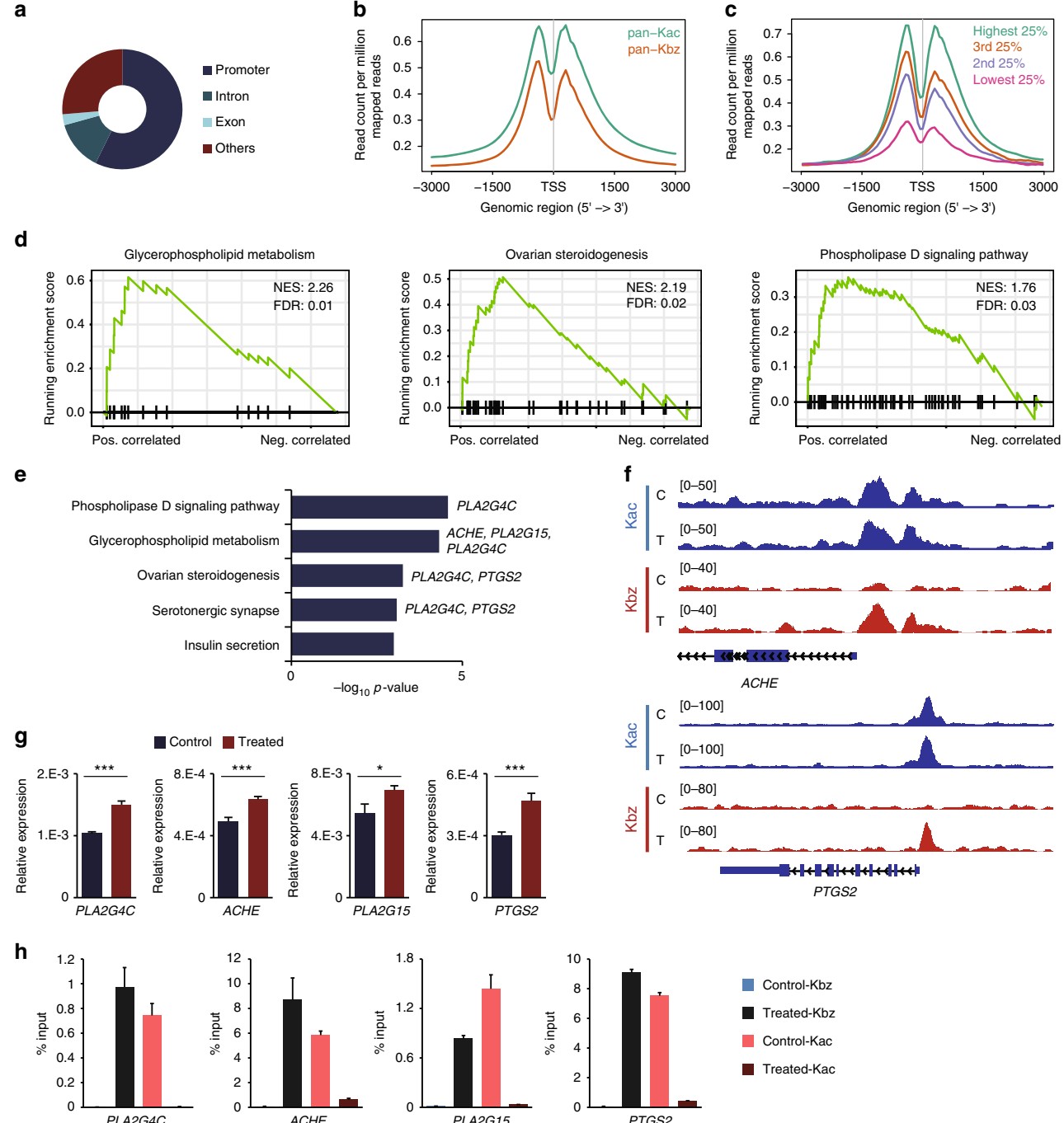

**Fig. 6** Genome-wide localization and physiological relevance of histone $K_{bz}$ in HepG2 cells. **a** Genome-wide distribution of histone $K_{bz}$ in SB-treated HepG2 cells. **b** Curves showing average profile of histone $K_{bz}$ and $K_{ac}$ ChIP-seq read counts (per million mapped) around all known TSSs. **c** All genes were split into four equal groups based on their expression levels calculated from RNA-seq data. The average profiles of histone $K_{bz}$ ChIP-seq read counts (per million mapped) for each group were plotted at all known TSSs. **d** KEGG pathway analysis of RNA-seq data using GSEA. **e** KEGG pathway analysis of selected genes. The genes selected for validation in each pathway were labeled. **f** A representative snapshot showing the normalized ChIP-seq reads for $K_{bz}$ and $K_{ac}$ at *ACHE* and *PTGS2* genes (C: control, T: SB-treated). **g** RT-qPCR analysis of the control and SB-treated HepG2 cells. Relative expression is normalized to *ACTIN*. *$p < 0.05$, ***$p < 0.001$ ($n = 4$, unpaired Student's *t* test, values are expressed as mean ± s.d.) compared to control conditions. **h** qPCR analysis of $K_{bz}$ and $K_{ac}$ ChIP products from the control and SB-treated HepG2 cells ($n = 3$, values are expressed as mean ± s.d.)

## Methods

**Materials and reagents.** Unless otherwise noted, all chemical reagents were purchased from Sigma-Aldrich (St. Louis, MO). $D_5$-SB was from C/D/N Isotopes Inc., D-288. Antibodies were the following: anti-pan $K_{ac}$ (1:1000, PTM Biolabs, PTM-101), anti-H4K16$_{ac}$ (1:15,000; PTM Biolabs, PTM-122), anti-H3 (1:5000, Abcam, ab1791), anti-H4 (1:5000, Abcam, ab31830), anti-Flag (1:10,000; Sigma-Aldrich, F7425), anti-Tubulin (1:10,000; Abcam, ab6160), and anti-SIRT2 (1:1000, Abcam, 23886). Anti-pan $K_{bz}$ antibody (1:1000) was made by PTM Biolabs Inc. (Chicago, IL). MEFs derived from WT and *Sirt2* KO mice are kind gifts of Dr. Chu-

Xia Deng (NIH)[35]. HepG2 (HB-8065), RAW (TIB-71), and 293T (CRL-3216) cell lines were purchased from ATCC (www.atcc.org) and used without further authentication. No mycoplasma contamination was detected using a MycoAlert™ Mycoplasma Detection Kit (Lonza, LT07-118).

**Western blot analysis.** Protein extracts (20 μg of whole-cell proteins or 2–4 μg of histones) were fractionated by sodium dodecyl sulfate-polyacrylamide gel electrophoresis and transferred to a polyvinylidene difluoride membrane using a transfer

apparatus according to the manufacturer's protocols (Bio-Rad, Hercules, CA). After incubation with 3% bovine serum albumin (BSA) in TBST (10 mM Tris, pH 8.0, 150 mM NaCl, 0.5% Tween 20) for 1 h, the membrane was incubated with the indicated primary antibody (concentration is shown in "Materials and reagents" section) at 4 °C overnight. Then the membrane was washed three times (10 min for each) with TBST and incubated with a 1:20,000 dilution of horseradish peroxidase-conjugated anti-mouse or anti-rabbit antibodies at room temperature for 1 h. Next, the membrane was washed three times (10 min for each) with TBST and developed with enhanced chemiluminescence detection system (Thermo Scientific, Rockford, IL) according to the manufacturer's protocols. Uncropped western blots are shown in Supplementary Figures 7–9.

**Chemical propionylation.** Two hundred fifty µg of histone sample was dissolved in 250 µL of 0.1 M $NH_4HCO_3$ buffer (pH = 8.0), followed by adding 3.0 µL propionic anhydride and then adjusting pH to 8–9 with instant tunnel. After 1 h incubation at room temperature, we added another 3.0 µL of propionic anhydride and kept the pH at 8–9 by addition of 1 M NaOH. The mixture was incubated at room temperature for another 1 h and the reaction was blocked by adding 3.0 µL of 2-aminoethanol. The mixture was incubated for 15 min and digested with trypsin (histone: trypsin = 50:1) at 37 °C overnight.

**CoA extraction.** Cell culture media was quickly removed and the dish was placed on top of dry ice. Then 1 mL of extraction buffer (80% methanol/water, v/v) was immediately added and the dishes were transferred to a −80 °C freezer. The dishes were left at −80 °C for 15 min and transferred on top of dry ice. Next, cells were scraped into extraction solvent and the solution was centrifuged with the speed of $20,000 \times g$ at 4 °C for 10 min. The supernatant was dried in Speed Vac (Thermo-Fisher Scientific, San Jose, CA) for HPLC-MS/MS analysis.

**Immunoprecipitation.** The peptides in $NH_4HCO_3$ solution were incubated with prewashed pan anti-$K_{bz}$ beads (PTM Biolabs Inc., Chicago, IL) at 4 °C overnight with gentle shaking. After incubation, the beads were washed three times with NETN buffer (50 mM Tris pH 8.0, 100 mM NaCl, 1 mM EDTA, 0.5% NP40), twice with ETN buffer (50 mM Tris pH 8.0, 100 mM NaCl, 1 mM EDTA), and once with water. The bound peptides were eluted from the beads with 0.1% trifluoroacetic acid, and the eluted fractions were combined and vacuum-dried.

**Immunofluorescence staining.** HepG2 and RAW cells were seeded on coverslips before experiment. The cells on coverslips were washed twice with phosphate-buffered saline (PBS, 137 mM NaCl, 2.7 mM KCl, 10 mM $Na_2HPO_4$, and 2 mM $KH_2PO_4$) and fixed in 4% paraformaldehyde at room temperature for 15 min. After rinsing with PBS twice, the coverslips were incubated with 0.5% Triton X-100 at room temperature for 10 min, blocked with 3% BSA at room temperature for 60 min, and incubated with mouse pan anti-$K_{ac}$ (1:100) and rabbit pan anti-$K_{bz}$ (1:100) antibody mixture at 4 °C overnight. The coverslips were washed twice with PBST (PBS with 0.1% Tween 20), followed by incubation with Alexa Fluor 488-conjugated and Alexa Fluor 594-conjugated secondary antibodies (1:100, against rabbit and mouse, respectively) at room temperature for 60 min. Next, the cells were counterstained with 4,6-diamidino-2-phenylindole and mounted onto glass slides. Images were acquired with a Leica SP5 confocal microscope system.

**HDAC screening.** The reactions were performed in a final volume of 50 µL per well in a 96-well microplate. For each reaction, 0.5 µM of $K_{bz}$ peptide and HDACs (0.08 µM of HDAC1–11 or 0.25 µM of SIRT1–7) was added to reaction buffer (for HDAC1–11: 25 mM Tris pH 8, 130 mM NaCl, 3.0 mM KCl, 1 mM $MgCl_2$, 0.1% PEG8000, pH 8.0; for SIRT1–7: 20 mM Tris pH 8, 1 mM DTT, 1 mM $NAD^+$) in sequence. After 30 min incubation at 37 °C, the reactions were stopped by adding 50 µL of 200 mM HCl and 320 mM acetic acid in methanol. The samples were dried and analyzed by HPLC-MS/MS with a gradient of 5–90% HPLC buffer B (0.1% formic acid in acetonitrile, v/v) in buffer A (0.1% formic acid in water, v/v) at a flow rate of 900 nL/min over 15 min. Product quantification was based on their peak areas. The $K_{cat}$ and $K_m$ values were obtained by curve fitting the $1/V_{initial}$ versus $1/[S]$.

**Molecular docking.** Three-dimensional structure of SIRT2 (PDB ID 4Y6L) was downloaded from www.rcsb.org. Ligands were constructed with PyMol basing on the myristoyl peptide structure in PDB 4Y6L. Three residues of the ligand peptide, Arg-Lys-Ser, were kept, in which lysine was modified by different acylation groups. All the ligands were energy minimized using Chimera[36]. Receptor and ligand files for docking simulation were prepared using Autodock tools[37] (version 1.5.6). AutoDock vina[38] (version 1.1.2) program was employed for docking between protein and ligand information along with grid box properties in the configuration file. Grid center was designated at $x = 6.866$, $y = −13.796$, and $z = 39.416$, and the box size was set at $x = 30$, $y = 16$, and $z = 16$. Each docking simulation was run with the number of modes set to 9 and energy range set to 4. The docking results were manually checked and the pose with the lowest energy of binding was extracted. Binding detail of each ligand was further analyzed with LigPlot+ 1.4 program.

**HPLC-MS/MS analysis.** Histone peptide samples were dissolved in 2.5 µL of HPLC solvent A (0.1% formic acid in water, v/v) and loaded onto an in-house packed capillary $C_{18}$ column (10 cm length×75 µm ID, 3 µm particle size, Dr. Maisch GmbH, Ammerbuch, Germany), which was connected to EASY-nLC 1000 UHPLC system (ThermoFisher Scientific, San Jose, CA). Peptides were separated with a gradient of 5–90% HPLC buffer B (0.1% formic acid in acetonitrile, v/v) in solvent A at a flow rate of 200 nL/min over 60 min. The eluted peptides were analyzed by a Q Exactive mass spectrometer (ThermoFisher Scientific, San Jose, CA) using a nano-spray source. Full mass scans were acquired in the $m/z$ range of 300−1400 with a mass resolution of 70,000 at $m/z$ 200. The 15 most intensive ions were fragmented with 27% normalized collision energy and tandem mass spectra were acquired with a mass resolution of 17,500 at $m/z$ 200.

The extracted CoA samples were separated with 90% HPLC buffer B (5 mM $NH_4OAc$ in 95/5 $ACN/H_2O$, v/v) in buffer A (5 mM $NH_4OAc$) at a flow rate of 900 nL/min over 20 min. The eluted molecules were analyzed by an Orbitrap Velos mass spectrometer (ThermoFisher Scientific, San Jose, CA) with a targeted MS/MS method. Full mass scans were acquired with a resolution of 30,000 at $m/z$ 400; isolation width of precursor ions was set at $m/z$ 3.0, and high-energy collision dissociation was set at 35%[39].

**Protein sequence database searching.** Unrestrictive identification of PTMs was performed with PTMap algorithm. Mass shift of potential protein modifications ranged from −50 to +200 Da. Maximum missing cleavage was set at 3, and mass tolerance was set at ±0.01 Da for precursor ions and ±0.5 Da for MS/MS. Restrictive identification was performed with the Mascot search engine (Matrix Science, London, UK) against UniProt Human protein database (88,277 entries, http://www.uniprot.org). The following parameters were used during sequence alignment: methionine oxidation, protein N-terminal acetylation, lysine acetylation, lysine mono-/di-/tri-methylation, arginine mono-/di-methylation, and lysine benzoylation were specified as variable modifications. Maximum missing cleavage was set at 4, and mass tolerance was set at 10 ppm for precursor ions and ±0.05 Da for MS/MS. Propionylated histone samples for quantification were searched using MaxQuant v1.3.0.5 with integrated Andromeda search engine. Tandem mass spectra were searched against UniProt Human protein database (88,277 entries, http://www.uniprot.org) concatenated with reverse decoy database. Trypsin/P was specified as cleavage enzyme allowing up to four missing cleavages. Methionine oxidation, protein N-terminal acetylation, lysine acetylation, lysine mono-/di-/tri-methylation, arginine mono-/di-methylation, and lysine benzoylation were specified as variable modifications. FDR thresholds for protein, peptide, and modification site were specified at 1%. $K_{bz}$ identified on peptides from reverse or contaminant protein sequences and peptides with Andromeda score <40 were removed. All the $K_{bz}$ site ratios were normalized by the quantified protein expression levels.

**ChIP-seq analyses.** HepG2 cells were grown in full media with or without SB treatment (10 mM for 24 h). Chromatin was prepared by MNase digestion[40]. For each immunoprecipitation, 3 µg of pan anti-$K_{bz}$ or anti-$K_{ac}$ antibody were incubated with 30 µg chromatin overnight at 4 °C. ChIP-seq libraries were prepared following the TruSEq Chip-SEQ Kit (Illumina, San Diego, CA) as per the manufacturer's instruction. The libraries were sequenced with 50 bp single read sequencing on Illumina HiSeq 2500 machine at the University of Chicago Genomics Core as per the manufacturer's protocols. ChIP-seq reads were mapped to reference genome of Illumina iGenomes UCSC hg38 using Bowtie[41] (version 2.2.6), and only the uniquely mapped reads were retained. SAMtools[42] (version 0.1.19) was then used to convert files to bam format, sort, and remove PCR duplicates. Ngs.plot[43] (version 2.61) was used to generate average profiles of ChIP-seq reads. Peaks were called using MACS[44] (version 2.1.1) with FDR = 0.01, and data was visualized using IGV[45] (version 2.4). The differential binding analysis and KEGG Pathway analysis was implemented using DiffBind[46] and clusterProfiler[47] packages, respectively.

**RNA-seq analyses.** Total RNAs were extracted from control and SB-treated (10 mM for 24 h) HepG2 cells using the RNeasy Mini Kit (QIAGEN Inc, Valencia, CA). Four biological replicates were performed for each studied condition. The sequencing libraries were prepared using the TruSeq Stranded Total Sample Preparation Kit (Illumina, San Diego, CA) as per the manufacturer's instruction. The libraries were sequenced with 50 bp single read sequencing on Illumina HiSeq 2500 machine at the University of Chicago Genomics Core as per the manufacturer's protocols. RNA-seq reads were mapped to reference genome of Illumina iGenomes UCSC hg38 using TopHat[48] version 2.1.0. Mapped reads were summarized for each gene using featureCounts[49] version 1.5.0. Differential expression analysis was implemented using edgeR[50] version 3.16.5. Only the genes of which counts per million is larger than one in at least two samples were kept and the library sizes across samples were normalized using the TMM method in the edgeR package. GSEA analysis of KEGG pathway was implemented using the clusterProfiler[47] package.

**Primer sequences for ChIP-qPCR and RT-qPCR.** ChIP experiments for histone $K_{bz}$ and $K_{ac}$ were carried out according to the instructions from the ChIP-IT® High

Sensitivity Kit (Active Motif, Carlsbad, CA). ChIP products and input were analyzed by qPCR using SYBR® Select Master Mix (Life Technologies, Thermo Fisher Scientific Inc., Waltham, MA) and StepOnePlus Real-Time PCR system (Applied Biosystems, Thermo Fisher Scientific Inc., Waltham, MA). The sequences for ChIP-qPCR primers with forward followed by reverse are: *PLA2G4C* (GTGTTTCCTCCTGGTCCTGA/CTGAAAAAGCTGGAGCAACC), *ACHE* (GGGATCGCTAGTGGAAATGA/GTCCTGCCTTCTCAGGTGTC), *PLA2G15* (GCAGAGTTACAGGGGCTGAC/AGGAGCTGACCAGGACCTTT), and *PTGS2* (TCCCTCCTCTCCCCTTAAAA/CTGGGTTTCCGATTTTCTCA).

Total RNAs were extracted using the RNeasy Mini Kit (QIAGEN Inc, Valencia, CA). cDNA was prepared using RevertAid First Strand cDNA Synthesis Kit (Thermo Fisher Scientific Inc., Waltham, MA) following the manufacture's protocol and analyzed by qPCR using SYBR® Select Master Mix (Life Technologies, Thermo Fisher Scientific Inc., Waltham, MA) and StepOnePlus Real-Time PCR system (Applied Biosystems, Thermo Fisher Scientific Inc., Waltham, MA). The sequences for RT-qPCR primer pairs with forward followed by reverse are: *PLA2G4C* (TGGGCAATATCTTCTCTCTAC/GGTAAATCGATGTTTCAGGTC), *ACHE* (CTGTGGTAGATGGAGACTTC/CCCCGTAAACCAGAAAATAC), *PLA2G15* (GAAAGCTACTTCACAATCTGG/CAGCCTGATATTGTCAATCC), *PTGS2* (AAGCAGGCTAATACTGATAGG/TGTTGAAAAGTAGTTCTGGG), and *ACTIN* (ACCTTCTACAATGAGCTGCG/CTGGATGGCTACGTACATGG).

**Data availability**. The mass spectrometric data have been deposited to the ProteomeXchange Consortium with the dataset identifier PXD010332. The sequencing datasets reported in this paper have been submitted to the Gene Expression Omnibus database and are available under accession no. GSE108470.

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

## Acknowledgements

This project was supported by NIH grants (DK107868 and GM115961) to Y.Z. and National Center for Advancing Translational Sciences of the National Institutes of Health through Grant Number UL1 TR000430.

## Author contributions

Y.Z. and H.H. conceived and designed the experiments. H.H. performed the mass spectrometric analysis, biology experiments, western blot, modeling study, enzyme screening and analyzed ChIP-seq/RNA-seq data. D.Z. performed immunoprecipitation, ChIP, and qPCR experiments. M.P. was involved in western blot analysis. Y.W., Z.H., Y.G.Z. and Q.H. were involved in the enzyme screening. All authors discussed the data. H.H. and Y.Z. wrote the paper.

## Additional information

**Competing interests:** Y.Z. is a member of the scientific advisory board of PTM Biolabs Inc. (Chicago, IL). The remaining authors declare no competing interests.

