## [Peer Review File · Nature Communications]

Reviewers' comments:

Reviewer #1 (Remarks to the Author):

In this study, Huang and colleagues report on the new posttranslational modification (PTM) on histone H3, benzoylation of lysine. The authors used a combination of MS, biochemical and enzymatic assays, as well as in vivo ChIP-seq and genome mapping experiments to convincingly demonstrate the existence of this novel PTM. The manuscript describes exciting and novel findings and contains excellent quality data. I particularly like the idea that this mark is produced in large amount when cells are subjected to a chemical that is widely used to preserve human food. Here we actually deal with the effect of environment and may witness a very straightforward and easy understandable consequence of altering the human epigenome. The manuscript reads exceptionally well, and I enthusiastically support this work. A minor suggestion: abstract, unclear word 'non-conical'. Fig 5e-g, I would suggest moving panels (e and g) to a supplement and delete the panel (f)- this level details should be shown only for the actual structure but not in silico modeling. Best regards, Tatiana Kutateladze

Reviewer #2 (Remarks to the Author):

In this manuscript, Huang et al. report the discovery of lysine benzoylation (Kbz), a previously unidentified histone post-translational modification (PTM). The authors utilize HPLC-MS/MS analysis of core histones to identify +104 mass adducts that they eventually attribute to benzoylation of the epsilon-amino group on lysine side chains. This mark primarily occurs on the disordered N-terminal tails of histones, whereas acetylation (Kac) and crotonylation (Kcr) are found on lysines throughout histone sequences. Next, a pan-Kbz antibody is developed to study the Kbz mark in cell-based assays. While the levels of this mark are relatively low in the mammalian cell lines tested, addition of 5 mM sodium benzoate (SB) is able to stimulate accumulation of Kbz on histones. The authors are also able to detect the presence of benzoyl-CoA upon addition of SB and show that Kac levels are unaffected by this treatment. To unveil enzymes with the potential to regulate Kbz, a screen of known HDACs is performed that shows only recombinant SIRT2 has significant debenzoylase activity. No acyltransferases are identified in the manuscript (the promiscuous acyltransferase p300 was tested and did not show activity). SIRT2 activity is then confirmed in mammalian cells using knockout and overexpression experiments. Finally, genome-wide mapping via ChIP-seq shows a majority of histone Kbz occurs near TSS and that distribution is nearly identical to Kac. To analyze the effect of Kbz on gene expression, RNA-seq data is collected and combined with gene set enrichment analysis. This experiment identifies several pathways that are positively correlated with Kbz levels.

While the identification of Kbz on histones is of potential interest, it is much less clear whether this mark exists at physiologically relevant levels in cells (i.e. accumulates to a point where there is a transcriptional response). The SB concentrations used in this study very likely do not represent levels that exist in nature or are caused by diet (as suggested) and the transcriptional responses (even in the presence of extremely high and biologically irrelevant SB concentrations) are very modest. Many of the conclusions regarding biological relevance are not well supported by experimental evidence and therefore this manuscript, at least in its current form, is not at level one would wish for publication in Nature Communications. Explanations for these comments are below.

My major issues with this manuscript are as follows:

1. The cell-based assays including ChIP, RNA-seq, and SIRT2 knockout/overexpression all require the addition of 5 mM SB. The authors argue that this concentration is biologically relevant because it is within the range of SB allowed as a food preservative. This is not a valid point as the true physiologically relevant value would be the concentration of SB in the body/blood plasma after

consumption of such food (which would likely be several orders of magnitude lower than the 5 mM in food). Furthermore, the authors ignore potential buffering molecules in the blood plasma/in vivo environment, which would likely further diminish reactive SB species. With the exception of hyperammonaemia treatments, the concentrations of SB used in this study will never exist due to food consumption. It would be more appropriate to measure SB concentrations in people/mice who consume a high SB-containing diet.

2. The authors limit their experiments to Kbz on histones substrates and the ensuing transcriptional effect of this PTM. Conceivably (like other lysine acylation events) this PTM could be occurring on lysine side chains throughout the proteome, which could have many functional consequences. There is no rationale provided for why the study was restricted to histone substrates. In fact, the transcriptional responses of 'activated genes' (Fig. 6g) are quite modest (~1.2 to 1.3-fold) even when Kbz is highly upregulated at the corresponding promoters (Fig. 6h). This result suggests that if indeed SB causes 'harm to consumers' then histone Kbz/transcriptional effects are likely not the primary reasons. The authors should perform western blot and HPLC-MS/MS analysis on whole cell lysates (as well as on IP enriched lysate samples) to get a more clear picture of Kbz substrates throughout the proteome.

3. While it is clear that SIRT2 can remove this mark in recombinant assays and in live cells, it is not valid to imply that it is the only/primary enzyme responsible for this activity. Many HDACs function in large complexes (HDAC1/2 in particular) and do not show activity when isolated in recombinant assays. Because the authors strongly implicate SIRT2 as a regulator of Kbz (it is in the title), more experiments are necessary on this front. The authors should treat these cells with HDAC inhibitors (ex. SAHA) and show that levels of Kac increase but Kbz is unaffected. This would rule out Zn-dependent HDACs as potential Kbz regulators. Furthermore, the authors should knock out/overexpress another SIRT enzyme to show that SIRT2 is indeed selective over other sirtuins.

4. Additionally, regarding enzymatic regulation of this mark, the authors state the p300 is unable to catalyze the installation of Kbz. It is quite likely that some (if not all) of the observed Kbz is non-enzymatic and caused by the use of relatively high concentrations of SB in the assays (a common phenomena that is also observed with Kac in the presence of high concentrations of AcCoA). This idea should be discussed as it has implications for regulation (or lack thereof) of Kbz installation and its biological relevance.

5. While Kbz appears to occur at active TSS (Fig. 6b/c), the transcriptional response of target genes is very modest. Figures 6f and 6h show substantial (10-fold or more) increases in Kbz at gene promoters (ex. ACHE, PTGS2) upon treatment with SB. However, the corresponding transcriptional activation of these genes following SB treatment is modest (~1.2 or 1.3-fold). This result suggests that the effect of histone Kbz on transcription is inconsequential in a physiologically relevant context (where SB concentrations/fluctuations will be far lower). It is also not clear that Kbz is responsible for the modest activation that is observed for these genes. It is possible that addition of 5 mM SB to cells causes metabolic stress that is mildly activating these genes through some alternative pathway – i.e. an indirect effect.

6. The authors continuously stress that Kac levels are 'stable' or 'decrease slightly' (lines 140-143, 159-161) as measured by western blot and immunofluorescence. This is consistent with the ChIP-Seq data presented in Fig. 6f which shows that Kac in control and treated samples is unchanged at promoter regions of the AHCE and PTGS2 genes. However, Fig. 6h shows the exact opposite result; acetylation at ACHE and PTGS2 promoters drops dramatically (6 to 8-fold reduction) in response to SB treatment. These results are completely contradictory despite the fact that they are essentially the same assay (ChIP-Seq vs ChIP-qPCR) analyzing the same genes (ACHE and PTGS2). It does not make sense that acetylation would appear unchanged at these sites based on ChIP-Seq data and that acetylation would be appear dramatically down-regulated at these same sites based on ChIP-qPCR data. This must be reconciled prior to publication in any peer-reviewed journal.

Minor points:

1. What are the peptide sequences used for the dot blot validation?
2. In Fig 5d, why don't you detect WT Sirt2 before overexpression in the HEK293T cells. It is a ubiquitously expressed enzyme that is present in 293 cells.
3. How were Sirt2 KO cell lines generated?
4. In Fig 6g the transcriptional effects are quite modest and the fold-increase should be included in text for clarity. Also, the figure should include statistical significance (p-values) on bar graphs.

Reviewer #3 (Remarks to the Author):

The MS from Huang and colleagues report on lysine benzoylation, a new histone lysine acylation modification. Recent data show that a number of short-chain lysine acylations are common in histones. These modifications apparently contribute to the regulation of chromatin structure and gene expression. There is therefore considerable interest in understanding how such histone lysine acylations are deposited and removed from chromatin substrate. The authors find that lysine benzoylation occurs in cells treated with sodium benzoate (SB), which appears to result in increased cellular benzoylCoA concentrations and increased histone benzoylation. Huang et al. map the histone sites containing this modification and analyze the impact of SB on the genome wide distribution of this modification. As other histone acylations, the authors find that benzoylation occurs mostly around the transcriptional start sites, congruent with the model that this modification influences genome expression. The authors also report experiments that demonstrate that SIRT2 can remove the Kbz modification from substrates, apparently due to a unique active site architecture. However it remains unclear how this modification is deposited.

This information advances our understanding of histone PTM modifications and is worthy of publication in Nature Communications. As indicated below, one major remaining question concerns how prominent this PTM is in comparison to other histone acylations.

Major Comments

1. Fig. 3: Data are obtained from cells that are treated with 5 mM SB. What about non-treated cells? In Fig. 2d they show that they obtain robust benzoylation signal in different untreated cell lines. It would be important to know for the field how common this new PTM is in comparison to eg. acetylation, especially in untreated cells. Is it only a minor fraction? Do the modifications reported in Fig. 2-3 only arise due to treatment with SB? Unless the authors demonstrate directly that this PTM arises in non-treated cells, there are remaining doubts concerning physiological relevance/importance.
2. Fig. 3: The authors report that the KBz sites distribute in a similar pattern as compared to Kac and Kcr sites. Are these latter sites based on prior data from the literature or were these peptides measured? Can the authors derive a more quantitative comparison using the spectral counting method to determine the relative abundance of KBz vs. Kac and Kcr? What about cells that are not treated with SB?
3. It is unclear how the authors measured D5-BenzoylCoA production (Figure 4a). It looks like the relevant methods section is missing. As this involves HPLC-MS/MS analysis, one would expect that the authors should be able to detect eg. AcetylCoA as well. It would be interesting to know the relative peak areas of AcetylCoA vs. BenzoylCoA at the different concentrations of DB tested to get an estimate to the molar ratios of these AcylCoA variants. Eg. what about the 0mM control? These data could help to further rationalize Fig. 4b, where the authors show that Kbz increases as a function of SB concentration while Kac remains constant.

4. There are numerous grammatical errors throughout the text such as line 221: ‘...and plot CHIP-seq peak signals of Kbz on these genes’. Past tense must be used. Nature Communications should do a careful copyediting job if the MS is accepted (as, after revision, it should be).

Response to referees

Reviewers' comments:

Reviewer #1 (Remarks to the Author):

In this study, Huang and colleagues report on the new posttranslational modification (PTM) on histone H3, benzylation of lysine. The authors used a combination of MS, biochemical and enzymatic assays, as well as in vivo ChIP-seq and genome mapping experiments to convincingly demonstrate the existence of this novel PTM. The manuscript describes exciting and novel findings and contains excellent quality data. I particularly like the idea that this mark is produced in large amount when cells are subjected to a chemical that is widely used to preserve human food. Here we actually deal with the effect of environment and may witness a very straightforward and easy understandable consequence of altering the human epigenome. The manuscript reads exceptionally well, and I enthusiastically support this work. A minor suggestion: abstract, unclear word 'non-conical'. Fig 5e-g, I would suggest moving panels (e and g) to a supplement and delete the panel (f)- this level details should be shown only for the actual structure but not in silico modeling.

Response: We appreciate the positive comments from this referee. As suggested by the referee, we corrected the word 'non-conical' as 'non-canonical'. In addition, we deleted **Fig. 5f**, and moved **Fig. 5e** and **5g** to the updated **Supplemental Fig. 6c** and **6d**.

Reviewer #2 (Remarks to the Author):

In this manuscript, Huang et al. report the discovery of lysine benzylation (Kbz), a previously unidentified histone post-translational modification (PTM). The authors utilize HPLC-MS/MS analysis of core histones to identify +104 mass adducts that they eventually attribute to benzylation of the epsilon-amino group on lysine side chains. This mark primarily occurs on the disordered N-terminal tails of histones, whereas acetylation (Kac) and crotonylation (Kcr) are found on lysines throughout histone sequences. Next, a pan-Kbz antibody is developed to study the Kbz mark in cell-based assays. While the levels of this mark are relatively low in the mammalian cell lines tested, addition of 5 mM sodium benzoate (SB) is able to stimulate accumulation of Kbz on histones. The authors are also able to detect the presence of benzoyl-CoA upon addition of SB and show that Kac levels are unaffected by this treatment. To unveil enzymes with the potential to regulate Kbz, a screen of known HDACs is performed that shows only recombinant SIRT2 has significant debenzoylase activity. No acyltransferases are identified in the manuscript (the promiscuous acyltransferase p300 was tested and did not show activity). SIRT2 activity is then confirmed in mammalian cells using knockout and overexpression experiments. Finally, genome-wide mapping via ChIP-seq shows a majority of histone Kbz occurs near TSS and that distribution is nearly identical to Kac. To analyze the effect of Kbz on gene expression, RNA-seq data is collected and combined with gene set enrichment analysis. This experiment identifies several pathways that are positively correlated with Kbz levels.

While the identification of Kbz on histones is of potential interest, it is much less clear

whether this mark exists at physiologically relevant levels in cells (i.e. accumulates to a point where there is a transcriptional response). The SB concentrations used in this study very likely do not represent levels that exist in nature or are caused by diet (as suggested) and the transcriptional responses (even in the presence of extremely high and biologically irrelevant SB concentrations) are very modest. Many of the conclusions regarding biological relevance are not well supported by experimental evidence and therefore this manuscript, at least in its current form, is not at level one would wish for publication in Nature Communications. Explanations for these comments are below.

Response: We want to thank referee for pointing out this issue. Indeed, several lines of evidence suggest that Sodium Benzoate (SB) and Kbz can exist in significant levels.

First, SB and its metabolite Benzoyl-CoA are central intermediates in the degradation of a large number of aromatic growth substrates in both bacterial and human cells. Even in the absence of treatment, the plasma concentration of SB in healthy men ranges from 12.1 to 14.9 μM (*J. Nutr.* 2009, 139, 2309). Likewise, without SB treatment, we still detected several Kbz sites on core histones such as H3K23bz, H4K5bz, H4K8bz, H2AK9bz, and H2AK13bz, and their levels increased significantly in response to SB treatment (**Fig. 4d**). These results indicate that this histone mark exists in nature.

Second, importantly, the plasma SB concentration can be significantly elevated under certain physiological conditions. For example, the concentration of SB in plasma could be elevated to approximately 10 mM when patients with hyperammonaemia received intravenous SB (*J. Inherit. Metab.* 2000, 23, 129). In addition, oral administration of high dose SB increases the plasma SB concentration in healthy male volunteers into the range of 2-4 mM (*Eur. J. Clin. Pharmacol.* 1991, 41, 363). These concentrations of SB in plasma are similar to or even exceed the concentrations that we used for cell treatment, and are most likely associated with transcriptional responses.

Third, it is becoming increasingly accepted that acyl-CoAs can be generated at local intracellular compartments, directly contributing to the protein posttranslational modification (PTM) of proximal substrates (*Nature*. 2017, 552, 273; *Science*. 2009, 324, 1076). Therefore, the concentration of benzoyl-CoA generated in proximity to chromatin would likely exceed the average cellular concentrations and could facilitate the addition of Kbz to histones.

It should be noted that some important histone marks, e.g., H3K4me3, could not be detected by mass spectrometry in many cell lines (*J. Biol. Chem.*, 2007, 282, 7641). In contrast, multiple Kbz marks can easily be detected here even in cultured cells without SB treatment. This line of evidence again suggests that histone Kbz marks exist in significant levels in cultured cells.

In sum, the evidence suggests that the Kbz mark can exist at physiologically relevant levels in cells. We also added additional discussion and clarifications to the manuscript.

My major issues with this manuscript are as follows:

1. The cell-based assays including ChIP, RNA-seq, and SIRT2 knockout/overexpression all require the addition of 5 mM SB. The authors argue that this concentration is biologically relevant because it is within the range of SB allowed as a food preservative. This is not a

valid point as the true physiologically relevant value would be the concentration of SB in the body/blood plasma after consumption of such food (which would likely be several orders of magnitude lower than the 5 mM in food). Furthermore, the authors ignore potential buffering molecules in the blood plasma/in vivo environment, which would likely further diminish reactive SB species. With the exception of hyperammonaemia treatments, the concentrations of SB used in this study will never exist due to food consumption. It would be more appropriate to measure SB concentrations in people/mice who consume a high SB-containing diet.

Response: As discussed above, it has been reported that plasma concentrations of SB in humans increase in a dose-dependent manner following oral administration of SB (*Eur. J. Clin. Pharmacol.* 1991, 41, 363). In that study, high dose SB was orally administered to six healthy male volunteers and the concentration of SB in their plasma was determined. At 1h after administration, individual plasma SB concentrations reached ~2-4 mM, which is equivalent to the concentration used in our study. These results indicate that the concentration used in our study is physiologically relevant.

2. The authors limit their experiments to Kbz on histones substrates and the ensuing transcriptional effect of this PTM. Conceivably (like other lysine acylation events) this PTM could be occurring on lysine side chains throughout the proteome, which could have many functional consequences. There is no rationale provided for why the study was restricted to histone substrates. In fact, the transcriptional responses of 'activated genes' (Fig. 6g) are quite modest (~1.2 to 1.3-fold) even when Kbz is highly upregulated at the corresponding promoters (Fig. 6h). This result suggests that if indeed SB causes 'harm to consumers' then histone Kbz/transcriptional effects are likely not the primary reasons. The authors should perform western blot and HPLC-MS/MS analysis on whole cell lysates (as well as on IP enriched lysate samples) to get a more clear picture of Kbz substrates throughout the proteome.

Response: In this study, we discovered a previously undescribed histone mark, lysine benzoylation (Kbz), and identified one of its regulatory enzymes, Sirt2. In addition, we detected 18 Kbz substrate sites on core histones and demonstrated that histone Kbz marks are associated with gene regulation and have unique physiological relevance. Although many genes were not greatly changed (e.g. 30-60%), their regulation observed in our RNA-seq assay could be reproducibly corroborated by RT-qPCR experiments from independent biological replicate samples. We also would like to point out that it is not uncommon for meaningful gene expression to coincide with slight but reproducible changes. One of the many examples is shown in a recently published paper (*Mol. Cell.* 2015, 58, 203), in which crotonate responsive genes were upregulated by ~50%, comparable to our observations for benzoate-regulated genes. Therefore, our data clearly demonstrate that histone benzoylation is associated with benzoate regulated gene expression.

We agree with the reviewer that the Kbz mark may spread widely throughout the proteome and have important functions on non-histone proteins, and we added additional relevant discussion to the manuscript. However, the work proposed by this reviewer would merit its own independent story that is beyond the scope of this paper.

3. While it is clear that SIRT2 can remove this mark in recombinant assays and in live cells,

it is not valid to imply that it is the only/primary enzyme responsible for this activity. Many HDACs function in large complexes (HDAC1/2 in particular) and do not show activity when isolated in recombinant assays. Because the authors strongly implicate SIRT2 as a regulator of Kbz (it is in the title), more experiments are necessary on this front. The authors should treat these cells with HDAC inhibitors (ex. SAHA) and show that levels of Kac increase but Kbz is unaffected. This would rule out Zn-dependent HDACs as potential Kbz regulators. Furthermore, the authors should knock out/overexpress another SIRT enzyme to show that SIRT2 is indeed selective over other sirtuins.

Response: In this study, one of the Sirtuins, Sirt2, could remove Kbz in our *in vitro* screen. We confirmed its deacetylase activity *in vivo*. In contrast, HDACs 1-11 did not show deacetylase activity towards Kbz in our *in vitro* screen. To demonstrate that HDACs 1-11 do not show activity towards Kbz in cells, we followed the reviewer's suggestion; performed a western blot analysis to check the Kac/Kbz dynamics in response to the treatment of a class I/II HDAC inhibitor, Trichostatin A (TSA). The results are consistent with our screening. Neither HepG2 nor RAW core histone Kbz levels visibly changed upon treatment with TSA, even though the core histone Kac levels (positive control) did increase under the same conditions (**Supplementary Fig. 6b**).

We feel that the knockout and overexpression of the remaining 6 Sirtuins are unnecessary because our goal has been to identify a functional Kbz deacetylase. We have already well demonstrated that Sirt2 removes histone Kbz both *in vitro* and *in vivo*. It is beyond the scope of this paper to present all the negative *in vivo* data for the other Sirtuins.

4. Additionally, regarding enzymatic regulation of this mark, the authors state the p300 is unable to catalyze the installation of Kbz. It is quite likely that some (if not all) of the observed Kbz is non-enzymatic and caused by the use of relatively high concentrations of SB in the assays (a common phenomena that is also observed with Kac in the presence of high concentrations of AcCoA). This idea should be discussed as it has implications for regulation (or lack thereof) of Kbz installation and its biological relevance.

Response: We agree with the reviewer that some Kac sites are reported as non-enzymatic products. This kind of chemical reaction is likely the major mechanism for lysine acylation reactions in mitochondria where there are no known acyltransferases and pH is high (**Trends Biochem. Sci.** 2016, 41, 231). Abundant evidence indicates that enzyme-catalyzed reactions are mainly responsible for lysine acylations in cytosol and nuclei (**Chem. Rev.** 115, 2419; **Cell Discovery** 3, 17016; **Mol. Cell** 58, 203; **Cell** 154, 297; **Cell Res.** 27, 946). Likewise, we believe that the histone Kbz reported in this study is regulated by a certain unidentified transferase. Since we only have preliminary *in vitro* assay data for the transferases (including p300), which we did not show in the manuscript, we removed corresponding statement in the "Discussion" section to avoid any confusion.

5. While Kbz appears to occur at active TSS (Fig. 6b/c), the transcriptional response of target genes is very modest. Figures 6f and 6h show substantial (10-fold or more) increases in Kbz at gene promoters (ex. ACHE, PTGS2) upon treatment with SB. However, the corresponding transcriptional activation of these genes following SB treatment is modest (~1.2 or 1.3-fold). This result suggests that the effect of histone Kbz on transcription is inconsequential in a physiologically relevant context (where SB concentrations/fluctuations

will be far lower). It is also not clear that Kbz is responsible for the modest activation that is observed for these genes. It is possible that addition of 5 mM SB to cells causes metabolic stress that is mildly activating these genes through some alternative pathway – i.e. an indirect effect.

Response: Acetylation on core histones is well known to be associated with active gene expression. Benzoylation, structurally similar to acetylation but derived from benzoate and benzoyl-CoA, is also found to be positively correlated with gene expression in our analysis. Interestingly, under SB treatment, there exists a set of upregulated genes that are not accompanied by increases in histone acetylation (which actually decreased in many instances), but rather associate with increased histone benzoylation. We propose that these SB-responsive genes, which are not regulated by acetylation, could be explained by histone benzoylation as a mechanism. As described above, although many of these genes were not greatly changed (e.g. 30-60%), their up-regulation observed in our RNA-seq assay could be reproducibly corroborated by RT-qPCR experiments from independent biological replicate samples. It is not uncommon for meaningful gene expression to coincide with slight but reproducible changes. One of the many examples is shown in a recently published paper (*Mol. Cell.* 2015, 58, 203), in which crotonate responsive genes were upregulated by ~50%, comparable to our observations for benzoate-regulated genes. Therefore, our data clearly demonstrate that histone benzoylation is associated with benzoate regulated gene expression.

6. The authors continuously stress that Kac levels are ‘stable’ or ‘decrease slightly’ (lines 140-143, 159-161) as measured by western blot and immunofluorescence. This is consistent with the ChIP-Seq data presented in Fig. 6f which shows that Kac in control and treated samples is unchanged at promoter regions of the AHCE and PTGS2 genes. However, Fig. 6h shows the exact opposite result; acetylation at ACHE and PTGS2 promoters drops dramatically (6 to 8-fold reduction) in response to SB treatment. These results are completely contradictory despite the fact that they are essentially the same assay (ChIP-Seq vs ChIP-qPCR) analyzing the same genes (ACHE and PTGS2). It does not make sense that acetylation would appear unchanged at these sites based on ChIP-Seq data and that acetylation would appear dramatically down-regulated at these same sites based on ChIP-qPCR data. This must be reconciled prior to publication in any peer-reviewed journal.

Response: In fact, the histone Kac levels in our quantitative ChIP-seq analysis did decrease at promoters of ACHE and PTGS2 genes upon SB treatment (by 8% and 16%, respectively).

Several factors may lead to the fold-change difference of Kac between ChIP-seq and ChIP-qPCR. First, Kac peaks at promoter regions are very broad (e.g. 1700 bp for ACHE), while the ChIP-qPCR only evaluates a small local part (e.g. 92 bp for the ACHE case) of the whole peak. Therefore, the ChIP-qPCR result is unlikely to reflect the dynamics of the broad Kac peaks exactly, which is quite different from the ChIP-qPCR evaluation of the sharp and narrow peaks of transcriptional factors. In our ACHE case, although the whole Kac peak of SB treated sample decreased only slightly, the Kac level at the local region for ChIP-qPCR evaluation decreased greatly (**Fig R1a**). Therefore, it is not surprising that the Kac ChIP-qPCR result for ACHE showed a dramatic drop. Additionally, Kac ChIP-qPCR evaluation at other local regions in the ACHE and PTGS2 promoters (**Fig R1b**) showed that Kac level in these regions decreased in a dose-dependent manner upon SB treatment and the decrease of Kac is not as significant as the dynamics showed in **Fig 6h**.

Fig R1. ChIP-qPCR analysis of Kac in ACHE and PTGS2 promoters. (a) Local region of ACHE promoter for Kac ChIP-qPCR evaluation in **Fig 6h**. (b) Additional ChIP-qPCR evaluation of Kac peaks at ACHE and PTGS2 promoters. The sequences for ChIP-qPCR primers with forward followed by reverse are: *ACHE-a1* (GAAGTTAGCGCAAGGCCAAG / GAGTGTCCCACGTCACCTTT), *ACHE-a2* (CGGCAGTGGAACTTCTGGA / GCGTGGCCAATGAATGCTAG), *ACHE-a3* (CATCTGTGCCCACTGTCTCC / CTTCAACTAGTGCGGCCAGA), *PTGS2-a1* (CTGATCCCTCCCTCTCCTCC / CATCCAAGGCGATCAGTCCA), *PTGS2-a2* (GACAGACTGGGGCGAGTAAG / AGCTATGTATGTATGTGCTGCA).

Second, the different mechanisms underlying ChIP-seq and ChIP-qPCR may contribute to the discrepancies in fold changes observed in the two experiments; such differences were also happened elsewhere (e.g. **Genes Dev.** 2011, 25, 2480). It is worth emphasizing that the identification of differences between data sets becomes challenging when differences occur globally rather than at specific sites across the genome (**Cell Rep.** 2014, 9, 1163), as is the case for histone benzoylation.

Third, the differences in PCR amplifying cycles for ChIP-seq library preparation (9 cycles) and in ChIP-qPCR (40 cycles) experiments could also be a source of variance.

Finally, please note that although there are variances in fold-change between ChIP-seq and ChIP-qPCR results, the trends toward increasing or decreasing values are the same and reproducible.

Minor points:

1. What are the peptide sequences used for the dot blot validation?

Response: Thanks for the referee to point out this issue. Peptide libraries were used for the dot blot validation. Each peptide library contains 10 residues CXXXXKXXXX, where X is a mixture of 19 amino acids (excluding cysteine), C is cysteine, and the sixth residue is a modified lysine residue as indicated. This information was added to the legend of **Fig 2c**.

2. In Fig 5d, why don't you detect WT Sirt2 before overexpression in the HEK293T cells. It is a ubiquitously expressed enzyme that is present in 293 cells.

Response: We are sorry for the mistake. It should be “anti-Flag”, instead of “anti-Sirt2”. We have corrected it.

3. How were Sirt2 KO cell lines generated?

Response: Primary mouse embryonic fibroblasts (MEFs) derived from wild type and Sirt2 knockout mice are kind gifts of Dr. Chu-Xia Deng (NIH). The generation method was reported on *Cancer Cell* 2011, 20, 487. We added the cell lines information in “Materials and Reagents” section and cited the reference.

4. In Fig 6g the transcriptional effects are quite modest and the fold-increase should be included in text for clarity. Also, the figure should include statistical significance (p-values) on bar graphs.

Response: We thank the referee for this suggestion. The fold change was added in text and the statistical significance was included on the bar graphs.

Reviewer #3 (Remarks to the Author):

The MS from Huang and colleagues report on lysine benzoylation, a new histone lysine acylation modification. Recent data show that a number of short-chain lysine acylations are common in histones. These modifications apparently contribute to the regulation of chromatin structure and gene expression. There is therefore considerable interest in understanding how such histone lysine acylations are deposited and removed from chromatin substrate. The authors find that lysine benzoylation occurs in cells treated with sodium benzoate (SB), which appears to result in increased cellular benzoylCoA concentrations and increased histone benzoylation. Huang et al. map the histone sites containing this modification and analyze the impact of SB on the genome wide distribution of this modification. As other histone acylations, the authors find that benzoylation occurs mostly around the transcriptional start sites, congruent with the model that this modification influences genome expression. The authors also report experiments that demonstrate that SIRT2 can remove the Kbz modification from substrates, apparently due to a unique active site architecture. However it remains unclear how this modification is deposited. This information advances our understanding of histone PTM modifications and is worthy of publication in Nature Communications. As indicated below, one major remaining question concerns how prominent this PTM is in comparison to other histone acylations.

Response: We appreciate the positive comments of this referee.

Major Comments

1. Fig. 3: Data are obtained from cells that are treated with 5 mM SB. What about non-treated cells? In Fig. 2d they show that they obtain robust benzoylation signal in different untreated cell lines. It would be important to know for the field how common this new PTM is in comparison to eg. acetylation, especially in untreated cells. Is it only a minor fraction? Do the modifications reported in Fig. 2-3 only arise due to treatment with SB? Unless the authors demonstrate directly that this PTM arises in non-treated cells, there are remaining doubts concerning physiological relevance/importance.

Response: We can detect several Kbz sites on core histones by mass spectrometry in untreated cells, although the Kbz sites are not as common as Kac in this case. The detected Kbz sites in untreated HepG2 and RAW cells include H3K23bz, H4K5bz, H4K8bz, H2AK9bz, and H2AK13bz. Their levels increased significantly in response to SB treatment (**Fig. 4d**). Consistent with the mass spectrometry data, western blot analysis also detected Kbz modified histones in untreated HepG2 cells, mouse liver, and *Drosophila* S2 cells, indicating Kbz is an evolutionarily conserved histone mark in mammalian and insect cells.

2. Fig. 3: The authors report that the KBz sites distribute in a similar pattern as compared to Kac and Kcr sites. Are these latter sites based on prior data from the literature or were these peptides measured? Can the authors derive a more quantitative comparison using the spectral counting method to determine the relative abundance of KBz vs. Kac and Kcr? What about cells that are not treated with SB?

Response: The distribution of Kac and Kcr are based on our previously reported results (*Cell*, 2014, 159, 458; *Chem. Rev.* 2015, 115, 2376). Given that Kac is the most abundant acylation in cells, we compared the relative abundance of Kbz with Kac. To this end, we performed IP experiments using histones extracted from SB treated (5 mM for 24h) HepG2 or RAW cells. Pan anti-Kac and pan anti-Kbz antibodies were used to enrich Kac and Kbz peptides. Kac and Kbz abundances were quantified with a spectral counting method as suggested by the reviewer.

In either HepG2 or RAW cells, Kbz levels on some sites, such as H2AK5, H2AK8, H3K9, H3K14, and H4K5, are higher than the half of corresponding Kac levels, while H3K23bz and H4K8bz levels range from 12.5% to 38.9% of corresponding Kac levels (**Supplemental Fig. 5**). In addition, H4K12bz level is 70.8% of H4K12ac level in HepG2 cells, while this ratio decreased to 37.5% in RAW cells. In contrast, H3K18bz level is 21.4% of corresponding Kac level in HepG2 cells, while the ration increases to 71.4% in RAW cells.

Given that the quantified Kac levels only showed slight changes and the quantified Kbz levels increased 26-72 fold upon SB treatment (**Fig. 4d**), we can speculate that Kbz level is lower than Kac level by 1-2 orders of magnitude in untreated cells. Here we would like to point out that the higher level of acetylation does not necessarily abolish the functional roles of benzoylation. A lot of evidence has indicated that multiple PTMs can exist on the same residue and exert different functions, even though their abundances are vary greatly. For example, H3K9 can turn genes on by being acetylated, or silence them just as easily when tri-methylated (*Cell* 2007, 129, 823; *Genome Res.* 2007, 17, 691). Interestingly, H3K9 acetylation and tri-methylation levels are very different. In *Drosophila*, the abundance of H3K9 acetylation is below 1%, while the abundance of H3K9 tri-methylation is higher than 40% (*Mol. Cell.* 2015, 57, 559). Similar results were also observed in IMR90 lung fibroblast cells (*Methods*, 2015 90, 8). We would also like to point that Kbz levels are equivalent or close to Kac levels in many sites upon treatment of SB. Together, we believe Kbz play important and unique roles in cellular processes.

3. It is unclear how the authors measured D5-BenzoylCoA production (Figure 4a). It looks like the relevant methods section is missing. As this involves HPLC-MS/MS analysis, one would expect that the authors should be able to detect eg. AcetylCoA as well. It would be interesting to know the relative peak areas of AcetylCoA vs. BenzoylCoA at the different

concentrations of DB tested to get an estimate to the molar ratios of these AcylCoA variants. Eg. what about the 0mM control? These data could help to further rationalize Fig. 4b, where the authors show that Kbz increases as a function of SB concentration while Kac remains constant.

Response: We measured CoAs following a previously reported method (*Mol. Cell. Proteomics*. 2015, 14, 1489). A description of CoA extraction and analysis was added in the “Methods” section. Actually, we determined both acetyl-CoA and benzoyl-CoA in HepG2 cells at same time. As shown in updated **Fig. 4a**, the Ac-CoA has no obvious change when the cells were treated with 1 mM of SB, while it decreased ~14% and ~15% when the cells were treated with 5 mM and 10 mM SB, respectively. This result is consistent with the Kac western blot and the quantified Kbz and Kac site dynamics, in which core histone Kac levels slightly decreased upon treatment of 5mM SB.

4. There are numerous grammatical errors throughout the text such as line 221: ‘...and plot ChIP-seq peak signals of Kbz on these genes’. Past tense must be used. Nature Communications should do a careful copyediting job if the MS is accepted (as, after revision, it should be).

Response: We carefully checked the manuscript and corrected the grammatical errors. All the corrections are marked in red in the revised manuscript.

REVIEWERS' COMMENTS:

Reviewer #1 (Remarks to the Author):

The authors have well addressed all previous comments.

Reviewer #2 (Remarks to the Author):

The authors have done a nice job of addressing my concerns through significant textual changes and the addition of new data. In particular the Trichostatin A treatment makes the contention that SIRT2 is the major eraser much stronger. I now support acceptance of this manuscript.

Response to reviewers

REVIEWERS' COMMENTS:

Reviewer #1 (Remarks to the Author):

The authors have well addressed all previous comments.

Response: We appreciate the positive comment from this reviewer.

Reviewer #2 (Remarks to the Author):

The authors have done a nice job of addressing my concerns through significant textual changes and the addition of new data. In particular the Trichostatin A treatment makes the contention that SIRT2 is the major eraser much stronger. I now support acceptance of this manuscript.

Response: We appreciate the positive comments from this reviewer.